# p53N236S Activates Autophagy in Response to Hypoxic Stress Induced by DFO

**DOI:** 10.3390/genes13050763

**Published:** 2022-04-26

**Authors:** Kang Gao, Huanhuan Zong, Kailong Hou, Yanduo Zhang, Ruyi Zhang, Dan Zhao, Xin Guo, Ying Luo, Shuting Jia

**Affiliations:** 1Laboratory of Molecular Genetics of Aging and Tumor, Medical School, Kunming University of Science and Technology, Kunming 650500, China; gaokangwhj@163.com (K.G.); 20182136018@stu.kust.edu.cn (H.Z.); 20213136001@stu.kust.edu.cn (K.H.); 20212136016@stu.kust.edu.cn (Y.Z.); r.zhang@erasmusmc.nl (R.Z.); danzhao06@126.com (D.Z.); guoxin1996@foxmail.com (X.G.); 2Guizhou Provincial Key Laboratory of Pathogenesis and Drug Development on Common Chronic Diseases, School of Basic Medicine, Guizhou Medical University, Guiyang 550000, China; luoying@gmc.edu.cn

**Keywords:** p53N236S, autophagy, deferoxamine

## Abstract

Hypoxia can lead to stabilization of the tumor suppressor gene p53 and cell death. However, p53 mutations could promote cell survival in a hypoxic environment. In this study, we found that p53N236S (p53N239S in humans, hereinafter referred to as p53S) mutant mouse embryonic fibroblasts (MEFs) resistant to deferoxamine (DFO) mimic a hypoxic environment. Further, Western blot and flow cytometry showed reduced apoptosis in *p53^S/S^* cells compared to WT after DFO treatment, suggesting an antiapoptosis function of p53S mutation in response to hypoxia-mimetic DFO. Instead, *p53^S/S^* cells underwent autophagy in response to hypoxia stress presumably through inhibition of the AKT/mTOR pathway, and this process was coupled with nuclear translocation of p53S protein. To understand the relationship between autophagy and apoptosis in *p53^S/S^* cells in response to hypoxia, the autophagic inhibitor 3-MA was used to treat both WT and *p53^S/S^* cells after DFO exposure. Both apoptotic signaling and cell death were enhanced by autophagy inhibition in *p53^S/S^* cells. In addition, the mitochondrial membrane potential (MMP) and the ROS level results indicated that p53S might initiate mitophagy to clear up damaged mitochondria in response to hypoxic stress, thus increasing the proportion of intact mitochondria and maintaining cell survival. In conclusion, the p53S mutant activates autophagy instead of inducing an apoptotic process in response to hypoxia stress to protect cells from death.

## 1. Introduction

The modulation of oxygen concentration in vivo is a fundamental property of cell physiology. Low oxygen level (hypoxia) is a prominent feature of most solid tumors and generally correlates with poor prognosis, malignant phenotype, and resistance to therapy in human cancer [1,2]. Hypoxia stress reaction refers to a serial response when cells or tissues are under an oxygen-limited condition, including activation of the hypoxia-inducible factor (HIF) signaling cascade and an enhanced mitochondrial reactive oxygen species (ROS) production [3]. Some pathways that help the cell adapt to the situation or undergo cell death are activated as well, such as apoptosis, cell survival, and energy production via the p53/Mdm2, PI3k/Akt/mTOR, and glycolysis/TCA cycle pathways, respectively [4]. The transcription factor and tumor suppressor p53 has been shown to respond to hypoxia [5,6]. In general, under conditions of mild and/or transient hypoxia, p53 protein is kept at an uninduced level that supports increased glycolysis and decreased mitochondrial activity in concert with HIF-1α-mediated prosurvival pathways [7,8]. Under extended or severe hypoxia, however, p53 is strongly induced, HIF-1α levels generally decrease, and the p53-mediated apoptosis is processed [9,10].

Autophagy is a cellular self-digestion pathway involved in protein and organelle degradation in the lysosome. Many studies have demonstrated that activation of autophagy during hypoxia favors tumor cell survival [11,12,13]. However, the mechanisms of hypoxia-induced autophagy are not clear. Generally, it is dependent on the HIF-1α/AMPK pathway, and the finding that BNIP3, a HIF-1α downstream target gene, is essential to hypoxia-induced autophagy suggests one possible mechanism [14].

Recent studies have shown that p53 plays an important role in autophagy regulation which depends on its location in different intracellular compartments [15,16]. On one hand, nuclear p53 acts as a proautophagic factor by inducing the expression of autophagic target genes, such as DRAM [17], Sestrin1, Sestrin2 [18], ISG20L1, and AEN [19]. On the other hand, p53 inhibits the induction of autophagy in the cytoplasm by suppressing AMPK and activating the mTOR-dependent pathway [15]. However, the mechanism of autophagy inhibition by cytoplasmic p53 is not well understood [20]. Mutations in the TP53 gene are very common, occurring in approximately 50% of human tumor cells, leading to loss-of-function (LOF) and/or gain-of-function (GOF) mutations in the p53 protein [21]. The majority of TP53 gene mutations result in single amino-acid substitutions and mainly take place in the DNA-binding core domain of p53. These missense mutant p53 proteins commonly show increased stability and excessive accumulation compared to wild-type p53, which is thought to depend largely on the mutant p53 failing to regulate the p53-MDM2 feedback loop [22]. In addition, some chaperone proteins, such as Hsp90, Hsp70, and CHIP, also regulate the stability of mutant p53 proteins [23]. It has been shown that the nuclear accumulation of mutant p53 protein is associated with cell survival depending on autophagic activation. Activation of autophagy has been reported for the mutation p53 R175H, which is known to inhibit the nuclear and cytoplasmic effects of p53, as well as p53 NES (a p53 mutant that disrupts nuclear export signal) in HCT116 cells [15]. Thormas Naves et al. reported neuroblastoma p53 mutant cells (SKNBE(2c), expressing one allele with TP53 mutated at codon 135) engaging an autophagic pathway when treated with hypoxia-mimetic chemical CoCl_2_ [24].

To further understand how mutant p53 regulates hypoxia stress response, we generated missense mutant p53N236S (p53N239S in humans, hereinafter referred to as p53S) gene knock-in mouse embryonic fibroblasts (MEFs). The p53S mutant was found in three independent tumorigenic mouse alternative lengthening of telomeres (ALT) cell lines in our previous study [25]. We have demonstrated that the p53S mutant not only loses its tumor-suppressor function, but also gains some oncogenic functions, including enhanced migratory capability, generation of chromosome aberrations, increased tumorigenesis in vivo, and resistance to doxorubicin-induced apoptosis [25,26,27]. Here we reported that p53S induces cell autophagy and contributes to repressing apoptosis in response to hypoxia stress in a subcellular localization-dependent manner.

## 2. Materials and Methods

### 2.1. Hypoxia Treatment

Wild-type, *p53^S/S^*, and *p53^S/+^* MEFs were cultured for 4, 8, 16, or 24 h in a hypoxia modular incubator chamber (Billups-Rothenberg, Delmar, CA, USA) supplemented with an appropriate humidified gas mixture (94% N_2_, 5% CO_2_, and 1% O_2_) at 37 °C.

### 2.2. Cell Culture

The wild-type, *p53^S/S^*, and *p53^S/+^* MEFs were harvested and prepared from individual day 13.5 embryos from *p53^S/+^* mice breeding. The *p53^S/S^* knock-in mice were established by a commercial service (inGenious Targeting Laboratory Inc., Ronkonkoma, NY, USA). The details are described in another paper [27]. All cell lines were cultured in DMEM supplemented with 10% fetal bovine serum (Hyclone, Logan, UT, USA) in a 3% oxygen and 5% CO_2_ incubator at 37 °C. All mouse procedures were performed with the approval of the Animal Care and Use Committee of the Kunming University of Science & Technology (approval ID: M2018-0008).

### 2.3. Chemical Treatment

Deferoxamine (DFO, Sigma-Aldrich, St. Louis, MO, USA, D9533), chloroquine (CQ, Sigma-Aldrich, St. Louis, MO, USA, C6628), and 3-methyl adenine (3-MA, Sigma-Aldrich, St. Louis, MO, USA, 189490) were obtained from Sigma-Aldrich (Darmstadt, Germany). For DFO treatment, wild-type, *p53^S/S^*, and *p53^S/+^* MEFs were incubated with 200 μM DFO from 0 to 24 h before analysis. For 3-MA treatment, cells were first incubated with 10 mM 3-MA for 4 h before the addition of DFO. For CQ treatment, cells were first incubated with 25 μM CQ for 2 h and then treated with DFO for 16 h.

### 2.4. Electron Microscopy Analysis

Electron microscopy analysis was realized at the service of Kunming Medical University (China). Briefly, cells were fixed with 3.5% glutaraldehyde for 3 h, rinsed 3 times with 0.1 M phosphoric acid solution for 15 min, and postfixed with aqueous 1% osmium tetroxide for 3 h. After dehydration in ethanol and embedding in Epon, thin sections, stained with uranyl acetate and lead citrate, were examined with a JEM-1011 electron microscope (JEOL, Tokyo, Japan).

### 2.5. JC-10 Staining

JC-10 Mitochondrial Membrane Potential Assay Kit (Sangon Biotech, Shanghai, China) was used to measure the mitochondrial membrane potential change when wild-type, *p53^S/S^*, and *p53^S/+^* MEFs were treated with DFO. Briefly, after stress induction, cells were harvested and centrifuged to obtain 2–5 × 10^5^ cells per tube. Cells were resuspended in 500 μL of JC-10 dye-loading solution (with 2.5 μL 200 × JC-10) and incubated in the loading dye at room temperature or in a 37 °C, 5% CO_2_ incubator for 60 min, protected from light. The fluorescence intensity was monitored by using flow cytometry in the FL1 channel for the green fluorescent monomeric signal (in apoptotic cells) and the FL2 channel for the orange fluorescent aggregated signal (in healthy cells). Cytometric data were analyzed with FlowJo V10.6.2 (Burlingame, CA, USA).

### 2.6. Apoptosis Analysis and Flow Cytometry Assays

MEFs were grown overnight and treated or not with DFO for different durations. Cells were resuspended, washed with PBS, and centrifuged at 200× *g* for 3 min at 4 °C. Apoptotic cell death was measured using FITC Annexin V Apoptosis Detection Kit I (BD Biosciences, San Jose, CA, USA), according to the manufacturer’s protocol. Fluorescence intensity was measured using BD Accuri C6 Plus (BD Biosciences, San Jose, CA, USA). The “UR + LR” parts of the cytometry plots are considered as total apoptotic cells.

### 2.7. Immunofluorescence and Microscopy Analysis

Cells were spread onto coverslips and treated or not with DFO for different durations. After washing twice in PBS, cells were fixed with 3% paraformaldehyde—2% sucrose for 10 min at room temperature and permeabilized with 1% NP40 in PBS for 5 min at room temperature. After blocking in 5% BSA, coverslips were incubated with a p53 antibody (1:500, 9282, Cell Signaling, Beverly, MA, USA) overnight at 4 °C in a humid chamber and then with secondary Alexa Fluor 568-conjugated anti-rabbit IgG (Life Technologies, Waltham, MA, USA) for 1 h at room temperature (RT) in the dark. Slides were mounted in Vectashield Mounting Medium (Vector, H-1200). Images were captured on a Nikon Ti-E (Chiyoda-Ku, Tokyo, Japan) microscope using equal exposure times for all images.

### 2.8. ROS Detection

Cellular ROS were measured by flow cytometry using DCFH-DA as a fluorescent probe. Cells were treated or not with DFO for different durations. After washing twice in PBS, cells were incubated with 10 μM DCFH-DA for 30 min at 37 °C, washed, resuspended in PBS, and then analyzed using a BD Accuri C6 Plus (BD Biosciences, San Jose, CA, USA).

### 2.9. Western Blotting and Antibodies

Cells were lysed in RIPA buffer containing Protease Inhibitor Cocktail kit (Roche, Basel, Switzerland). Then, 20 μg of total protein was separated by SDS-PAGE and then transferred to a PVDF membrane. After blocking in 10% nonfat milk for 1 h at room temperature, membranes were incubated with primary antibodies overnight at 4 °C or 2 h at room temperature. The membranes were then incubated with horseradish peroxidase labeled secondary antibodies and visualized with ECL. The antibodies used and their sources are as follows: HIF-1α (1:500, MA1-516, Thermo Fisher, Waltham, MA, USA), p53Ab1 (1:1000, 9282, Cell Signaling, Beverly, MA, USA), phospho-p53 (Ser15) (1:500, 9284, Cell Signaling), LC-3 (1:5000, 2775, Cell Signaling), PARP (1:1000, 9542, Cell Signaling), caspase-3 (1:1000, 9662, Cell Signaling), GAPDH (1:1000, sc-32233, Santa Cruz, Dallas, TX, USA), Atg7 (1:1000, 8558, Cell Signaling), Atg12 (1:1000,4180, Cell Signaling), HSF1 (1:000, sc-17756, Santa Cruz), HSP70 (1:1000, Stressgen, Farmingdale, NY, USA), P-ser473 AKT (1:1000, 9271, Cell Signaling), AMPK (1:1000, 5831S, Cell Signaling), p-AMPK (1:1000, 5831, Cell Signaling), mTOR (1:1000, A2445, Cell Signaling), p-mTOR (1:1000, 2971, Cell Signaling), p62 (1:1000, 8025, Cell Signaling, Beverly, MA, USA), and anti-γ-tubulin (1:8000, t6557, Millipore, St. Louis, MO, USA).

### 2.10. Monodansylcadaverine (MDC) Staining

Cells were seeded on sterile coverslips in tissue culture plates. After DFO treatment for different durations, cells were washed once with PBS and stained with 0.05 mM MDC at 37 °C for 15 min. The occurrence of autophagic vacuoles was immediately observed under fluorescence microscopy (Nikon, Ti-E, Chiyoda-Ku, Tokyo, Japan).

### 2.11. Statistical Analysis

Data were expressed as the mean ± standard deviation (SD). Differences between two groups were analyzed using Student’s *t*-test, and values of *p* ≤ 0.05 were considered significant.

## 3. Results

### 3.1. p53N236S MEFs Show Resistance to DFO Mimicking a Hypoxic Microenvironment by Repressing Cell Apoptosis

To understand whether p53S can respond to hypoxia, we used the p53S homozygous (*p53^S/S^*) and heterozygous (*p53^S/+^*) transgenic MEFs. WT (control), *p53^S/S^*, and *p53^S/+^* cells were exposed to hypoxia (1% oxygen for 24 h) or treated with the hypoxia-mimetic DFO (200 μM, 4 h), and levels of HIF-1α, HSF1, and HSP70 were monitored by immunoblotting. The heat shock transcription factor 1 (HSF1) is classically activated by heat stress, reactive oxygen species (ROS), and hypoxia [28,29,30], leading to the induction of heat shock proteins (HSPs), such as HSP90, HSP70, and HSP27 [31]. HSPs are molecular chaperones that protect proteins from degradation, oxidative stress, hypoxia, and thermal stress [32]. Our data showed that the hypoxia stress induced HIF-1α and p53 expression, as well as p53S (Figure 1A). Moreover, the HSF1–HSP70 pathway was also upregulated in WT and p53S cells, suggesting a hypoxia response also in p53S cells (Figure 1A). Then, we tested if HIF-1α levels decrease over time during sustained hypoxia in a p53 mutant background. The results showed that the expression level of p53 protein, as well as Ser15 residue phosphorylation, gradually increased with the hypoxia treatment time, while the expression level of HIF-1α initially increased and then started to decrease at 16 h, indicating that the p53S mutant can also modulate the degradation of HIF-1α when the hypoxia condition becomes severe (Figure 1B). Similarly to previous reports (7), WT cells underwent apoptosis after a certain time of sustained hypoxia via a p53-dependent pathway. However, both *p53^S/S^* and *p53^S/+^* cells showed resistance to DFO treatment for 24 h (Figure 1C). This outcome was confirmed by the expression of cleaved PARP and cleaved caspase-3, which were increased in WT cells but dramatically decreased in both *p53^S/S^* and *p53^S/+^* cells after 16 h of 200 μM DFO treatment (Figure 1D). These results were confirmed by the flow cytometry analysis of apoptotic cells, which dropped to 18% at 16 h and 22% at 24 h in *p53^S/S^* cells compared with about 35% at 16 h and 45% at 24 h in both WT and *p53^S/+^* cells (Figure 1E,F). Thus, here we showed that DFO treatment induces a canonical p53-dependent apoptotic pathway in *p53* WT and *p53^S/+^* cells, whereas *p53^S/S^* cells are resistant to DFO-induced apoptosis.

### 3.2. DFO Simulates Hypoxia-Induced Autophagy in p53S MEFs

It has been reported that p53 mutant neuroblastoma cells engage an autophagic pathway when treated with the hypoxia-mimetic chemical CoCl_2_ [24]. To further understand the mechanism of p53 resistance to hypoxia stress, we examined the autophagy process in p53S MEFs in response to DFO treatment over time. It is noteworthy that both WT and *p53^S/+^* cells showed a decreased expression of ATG7 and ATG12 when treated with DFO in a time-dependent manner, whereas no significant change in ATG7 or ATG12 was observed in *p53^S/S^* cells after DFO treatment (Figure 2A). We also monitored the autophagy-related lipidated form of microtubule-associated protein 1A/1B-light chain 3 (LC3) and found that the LC3-II/LC3-I ratio dramatically increased in *p53^S/S^* cells in the presence of DFO after 16 h (Figure 2A). These results suggest that in response to DFO-induced hypoxia, the wild-type p53 protein in both WT and *p53^S/+^* cells suppresses autophagy, while at the same time promoting apoptosis. To evaluate autophagy in the presence of p53S, we measured the autophagy flux by using chloroquine (CQ), an inhibitor of autophagosome degradation. The *p53^S/S^* cells were firstly treated with CQ for 2 h, and then the autophagy substrate levels with or without DFO were measured. The p62 was degraded dramatically upon DFO treatment even in the presence of CQ, which indicated that DFO could promote autophagy activation in p53S mutation cells (Figure 2B). To further investigate the autophagy induced by DFO in *p53^S/S^* cells, we monitored the production of autophagic lysosomes in MEFs by using monodansylcadaverine (MDC) staining [33]. DFO treatment resulted in a more than 2-fold increase of MDC blue fluorescence staining in *p53^S/S^* cells (Figure 2C,D) compared with WT and *p53^S/+^* cells. These results reveal that DFO is able to induce autophagy in *p53^S/S^* cells, but not in WT and *p53^S/+^* cells.

Since p53 plays a dual role in the regulation of autophagy in a location-dependent manner [15,16], we sought to determine p53 localization in the absence or presence of hypoxia-induced stress. Immunofluorescence analysis showed that p53 was mainly located in the nucleus in the absence of stress in *p53^S/+^* and *p53^S/S^* cells, but it also can be detected in the cytoplasm (Figure 2G). Intriguingly, after 16 h of DFO treatment, the p53 protein translocated into the nucleus only in *p53^S/S^* cells. It is known indeed that cytoplasmic localization of p53 inhibits autophagy by suppressing AMPK and activating the mTO- dependent pathway; it has been demonstrated that in *p53*^−/−^ cells, AMPK and the AMPK substrates were hyperphosphorylated, whereas mTOR was hypophosphorylated [15]. Consistent with the translocation of p53S, the phosphorylation of AMPK was enhanced in *p53^S/S^* cells with DFO incubation but diminished in both *p53^S/+^* and WT cells under the same treatment. Moreover, the phosphorylation of mTOR and AKT was inhibited after DFO exposure in *p53^S/S^* cells, which probably leads to an activation of autophagy (Figure 2E,F). Taken together, these results suggest that the p53S mutant can translocate into the nucleus and promote cell autophagy in response to hypoxia stress.

### 3.3. p53S Suppresses DFO-Induced Apoptosis via Activation of the Autophagy Pathway

Recent advances have elucidated a dual role of autophagy associated with the regulation of apoptosis. In some cases, autophagy can block the apoptotic process and promote cell survival. However, autophagy or autophagy-relevant proteins may also help to induce apoptosis, which could aggravate cell death [34,35]. Nevertheless, the relationship between autophagy and apoptosis remains uncertain. To understand the interactions between autophagy and apoptosis in p53S cells in response to a low-oxygen environment, we used the autophagy inhibitor 3-methyladenine (3-MA) to determine whether the inhibition of autophagy is able to enhance hypoxia-induced apoptosis in vitro. The 3-MA treatment efficiently keeps the ratio of LC3-II/LC3-I at a low level (Figure 3A) in WT, *p53^S/+^,* and *p53^S/S^* cells. Then, we examined apoptotic activity in all three cells via PARP and caspase-3 cleavage and through annexin-V/PI staining. The cleaved PARP and caspase-3 levels were decreased in WT cells when autophagy was inhibited, as was the rate of the annexin-V/PI-stained apoptotic cells. However, the *p53^S/+^* cells showed only a slight decrease in the number of apoptotic cells when treated with 3-MA. Notably, the 3-MA treatment augmented the apoptosis of *p53^S/S^* cells in response to the hypoxia induced by DFO (Figure 3A,B). These results suggest that the apoptotic levels are enhanced by hypoxia alone, whereas apoptosis is suppressed after hypoxic treatment in combination with 3-MA in both WT and *p53^S/+^* cells. In contrast, in p53S mutant cells, autophagy inhibition increased the apoptosis levels, suggesting the autophagic process protects *p53^S/S^* cells against DFO stress.

We next tested whether the autophagy inhibition could change the location of p53S in hypoxic conditions using immunofluorescence staining. 3-MA treatment lead to the cytoplasmic location of p53S that translocated into the nucleus when treated with only DFO (Figure 3C). Thus, the regulation of autophagy in response to hypoxia is dependent on the location of p53S.

Our previous work showed that the p53S mutation gains tumorigenesis functions [25,27]. To evaluate whether the hypoxia response of p53S also represents a gain-of-function phenotype, we examined apoptotic and autophagic activity in *p53*^−/−^ cells treated with DFO and 3-MA. The results showed that DFO treatment induced apoptosis in about 22% of the *p53*^−/−^ cells, compared with about 15% of the *p53^S/S^* cells, indicating that the delayed apoptosis observed in p53S cells did not occur in p53-null cells after treatment with DFO (Figure 3B). This trend was consistent with Western blot results (Figure 3D). Moreover, 3-MA treatment can also increase the apoptosis level in *p53*^−/−^ cells, which suggests that the protective role of autophagy also exists in p53-null cells. Collectively, we showed that p53S gains a hypoxia-resistance function via activation of the autophagy pathway.

### 3.4. p53S Initiates Mitophagy to Clear Up Damaged Mitochondria in Response to Hypoxic Stress

Mitochondria are energy centers that produce adenosine triphosphate (ATP) through the process of cellular respiration. In addition, reactive oxygen species (ROS) are mainly generated in mitochondria. It has been reported that the ROS level increases under hypoxic stress, leading to mitochondrial oxidative damage [36]. Accumulation of dysfunctional mitochondria can be harmful to cells and organisms and drives a wide variety of pathogeneses [37]. It is believed that the selective degradation of damaged mitochondria is critical for cellular health [38]. Mitochondrial autophagy, or mitophagy, controls the mitochondrial quality, selectively degrading damaged mitochondria in order to maintain mitochondrial functions, and eliminates ROS-induced toxicity [39,40].

To investigate the influence of hypoxia on the mitophagy activity in p53S mutant cells, we firstly evaluated the mitochondrial transmembrane potential (ΔΨm) of p53S cells upon DFO treatment. To our surprise, the WT cells maintained a stable mitochondrial transmembrane potential both with and without DFO treatment (Figure 4A,B). A lower ΔΨm was detected in both *p53^S/+^* and *p53^S/S^* cells. Nevertheless, the transmembrane potential of mitochondria was significantly increased in *p53^S/S^* cells with the increase in DFO treatment time (Figure 4A,B). Moreover, flow cytometric analysis revealed that the levels of ROS were significantly lower in p53 dysfunctional cells, and DFO treatment failed to induce ROS in these cells. However, the 3-MA treatment partially recovered the ROS level in *p53*^−/−^, *p53^S/+^*, and *p53^S/S^* cells (Figure 4C). We speculate that hypoxia could restrict the production of ROS by upregulating the autophagy activity to remove damaged mitochondria. The electron micrographs revealed that autophagosomes, manifested as double-membrane vacuolar structures, containing damaged mitochondria or other organelles could be detected in *p53^S/+^* and *p53^S/S^* cells with DFO treatment, but not in WT cells (Figure 4D). These results indicate that in cells with a mutated or inactivated p53, mitophagy can be activated to protect mitochondria from ROS that are induced by hypoxia stress.

## 4. Discussion

Hypoxia commonly happens during the process of tumor growth. It is generally accepted that the severity of hypoxia varies between tumor types; on average, it is 1–2% O_2_ or below [41]. Recent studies have shown that hypoxia induces the activation of both autophagy and apoptosis in tumor cells [42]. On one hand, the oxygen deficiency increases tumor cell autophagy levels and promotes tumor cell survival [43,44]. However, in some cases, the activation of autophagy induced by hypoxia also leads to cell death. It has been shown that hypoxia induces autophagic cell death without inducing apoptosis in apoptosis-competent glioma and breast cancer cells [45]. On the other hand, lack of oxygen accelerates the induction of apoptosis in tumor cells. Accumulated DNA damage induced by hypoxia can stimulate the transactivation of genes encoding proapoptotic proteins, such as PUMA and NOXA, by p53 [46]. Moreover, autophagy and apoptosis can regulate each other under hypoxic conditions [42,47,48]. Generally, autophagy blocks the induction of apoptosis and inhibits the activation of the apoptosis-associated caspase, which could reduce cellular injury [44,49]. However, some studies suggest that the activation of autophagy may also promote apoptosis, which could aggravate cellular injury [50,51,52].

As previously described, p53 links autophagy and apoptosis in a complex, context-dependent manner, aiming to restore cellular and organismal homeostasis. The role of p53 as a regulator of apoptosis and autophagy has been extensively studied. Interestingly, as we mentioned before, p53 regulates both apoptosis and autophagy depending on its subcellular localization [15,16]. In the cytoplasm, p53 triggers apoptosis and blocks autophagy [24]. In the nucleus, p53 stimulates autophagy via its transactivation of downstream genes [17]. Under conditions of hypoxia and nutrient depletion, enhanced autophagy and antiapoptosis activity were observed in p53-deficient cancer cells [15,53], suggesting that dysfunctional p53 leads to stress resistance by induction of autophagy or suppression of apoptosis. In this study, we found that p53S, which lost its tumor suppressor function, activates autophagy in response to DFO-induced hypoxia. Similar to the previous report on p53 mutant SKNBE(2c) cells [24], p53S cells exhibited a delay in cell death when treated with a hypoxia-mimetic chemical (Figure 1D–F). However, DFO can induce a similar amount of apoptotic cells in WT, *p53^S/+^*, and *p53*-null MEFs, demonstrating that p53S may gain the function of regulating autophagy to block the apoptotic process and promote cell survival.

In our previous studies, we found that p53S accelerated cancer cell growth and metastasis [54]; it also gained unique functions in cross-talking with oncogenes and generating chromosome aberrations [25,27]. Moreover, the transcription and expression profile of p53S reveals that it regulates multiple pathways, including DNA damage response, estrogen response, and immune response [26]. However, p53S per se is not tumorigenic; it is likely to promote tumor growth by creating a tumorigenic microenvironment, for example by cancer-associated fibroblast (CAF) activation [54] and/or elimination of dysfunctional mitochondria in response to hypoxia (Figure 4D). Our data revealed a new aspect of the gain of function of p53S, which might act as a helper rather than a trigger in the tumorigenesis process.

Taken together, the results of this study showed that p53N236S MEFs engaged an autophagic pathway and the mutated p53S protein was mainly expressed in the nucleus in response to DFO stress, suggesting the nuclear mutated p53 protein benefits cell survival depending on the activation of autophagy, which could block the apoptotic process, remove damaged mitochondria, and possibly contribute to tumorigenesis.

## Figures and Tables

**Figure 1 genes-13-00763-f001:**
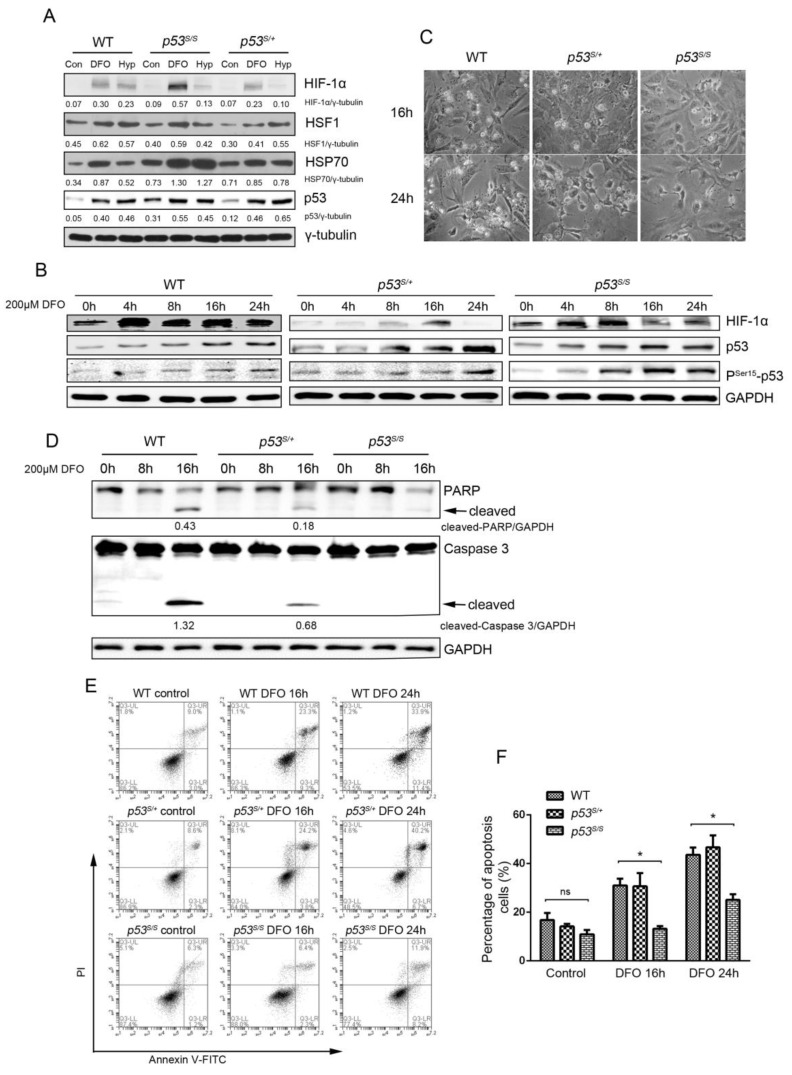
p53S cell response to hypoxia stress. (**A**) Expression of hypoxia-induced proteins was analyzed using Western blot with DFO treatment or hypoxia chamber culture. (**B**) Western blot analysis of HIF-1α, p53, and Pser15-p53 over a 24 h treatment with DFO. (**C**) The cell morphology was monitored using a phase-contrast microscope after a certain time of DFO treatment. (**D**) Western blot analysis of PARP and caspase-3 cleavage was carried out in WT, *p53^S/S^*, and *p53^S/+^* cells with DFO treatment. (**E**) Apoptosis induced by DFO treatment was monitored by flow cytometry analysis. (**F**) Quantification of E. Values represent means ± SD of at least three independent experiments. Statistical significance was evaluated using Student’s *t*-test. * *p* < 0.05. ns means no significance.

**Figure 2 genes-13-00763-f002:**
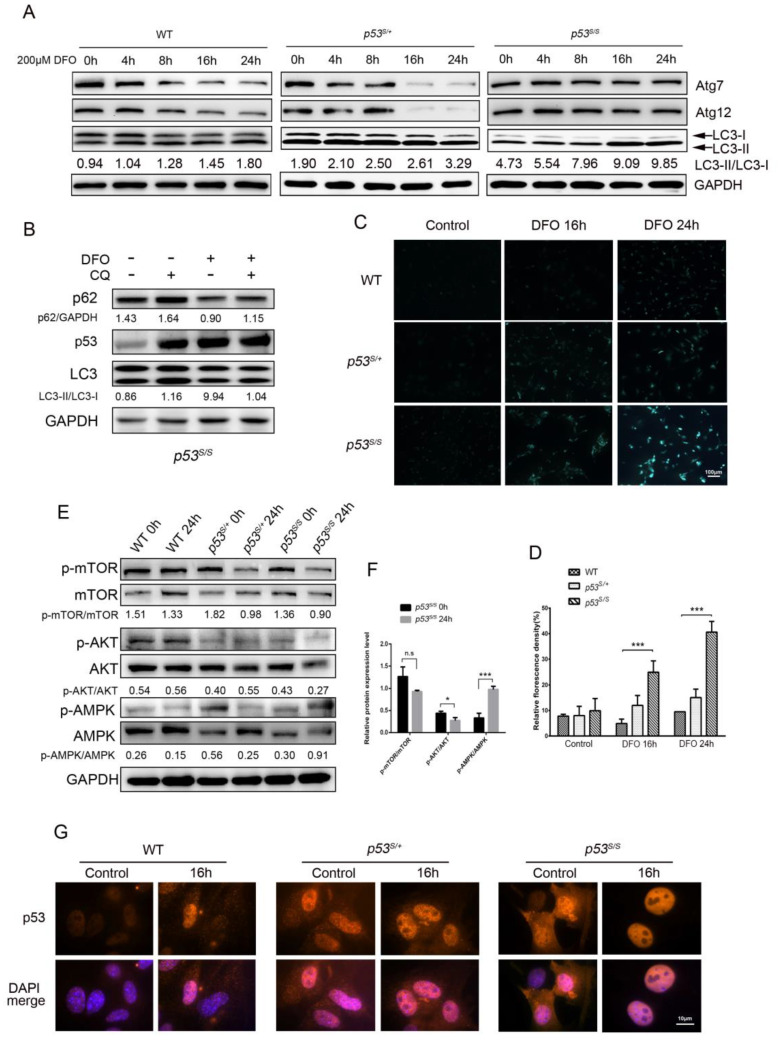
DFO induced autophagy in p53S MEFs. (**A**) Western blot analysis of Atg 7, Atg 12, and LC 3-I and -II in WT, *p53^S/+^*, and *p53^S/S^* cells with DFO treatment. (**B**) Western blot analysis of WT, *p53^S/+^*, and *p53^S/S^* cells with CQ and DFO treatment. (**C**) MDC staining of autophagic cells with DFO treatment. (**D**) Quantification of C. (**E**) Western blot analysis of AMPK and AKT/mTOR in WT, *p53^S/+^*, and *p53^S/S^* cells with/without 200 μM DFO exposure. (**F**) Quantification of the relative protein expression of p-mTOR, p-AKT, and p-AMPK in *p53^S/S^* cells. (**G**) The location of p53 was detected by immunofluorescence analysis (red) in cells treated with or without DFO. Nuclei were stained with DAPI (blue). Values represent means ± SD of at least three independent experiments. Statistical significance was evaluated using Student’s *t*-test. * *p* < 0.05, *** *p* < 0.001. ns means no significance.

**Figure 3 genes-13-00763-f003:**
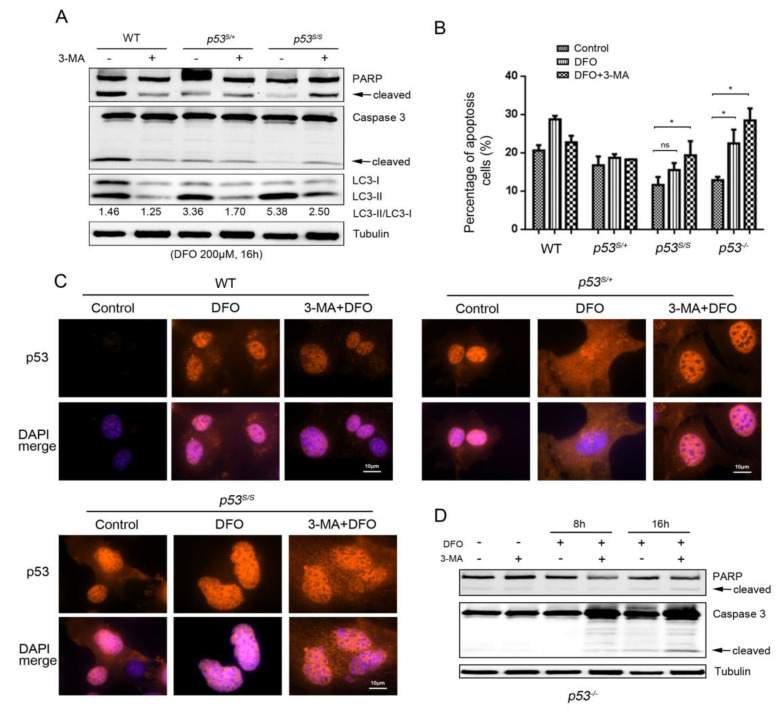
Inhibition of autophagy restored apoptosis activity in DFO-treated p53S MEFs. (**A**) Western blot analysis of autophagy and apoptosis proteins in cells treated with DFO and 3-MA. (**B**) Flow cytometry analysis of annexin-V/PI-stained apoptotic cells with DFO and 3-MA treatment. (**C**) The location of p53 was detected by immunofluorescence analysis (red) in cells treated with DFO and 3-MA. Nuclei were stained with DAPI (blue). (**D**) Western blot analysis of PARP and caspase-3 cleavage was carried out in *p53*^−/−^ cells with DFO and 3-MA treatment. Values represent means ± SD of at least three independent experiments. Statistical significance was evaluated using Student’s *t*-test. * *p* < 0.05. ns means no significance.

**Figure 4 genes-13-00763-f004:**
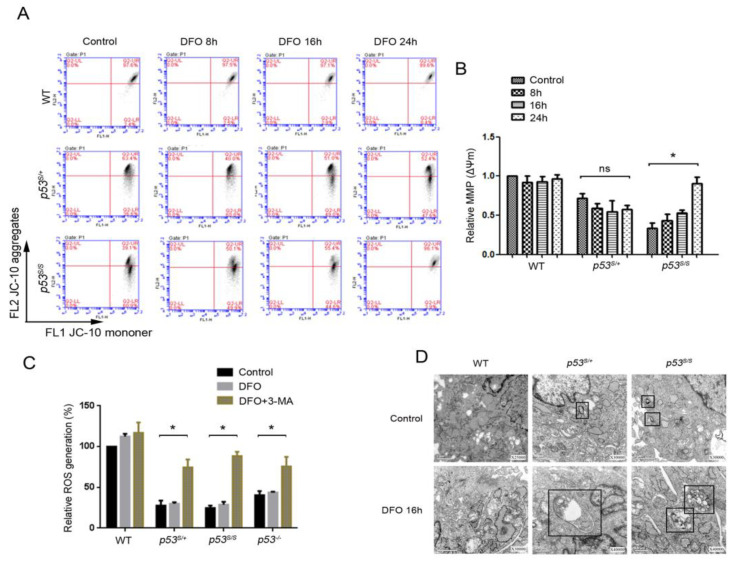
p53S initiates mitophagy to promote cell survival. (**A**) Mitochondrial transmembrane potential (MMP) was evaluated by JC-10 staining of DFO-treated cells. (**B**) Quantification of A. (mean of 3 separate experiments, * *p* < 0.05). The MMP of each sample was normalized by WT control. (**C**) Cellular ROS in DFO- and 3-MA-treated cells were measured by flow cytometry. (**D**) Representative electron micrographs of cells treated with DFO for 16 h. Double-membraned autophagosomes that contain cellular material are shown in the black frames. Values represent means ± SD of at least three independent experiments. Statistical significance was evaluated using Student’s *t*-test. * *p* < 0.05. ns means no significance.

## Data Availability

Not applicable.

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
