# Peer review of "p53N236S Activates Autophagy in Response to Hypoxic Stress Induced by DFO"

_genes, 2022, doi:10.3390/genes13050763_

Round 1

Reviewer 1 Report

Low cellular oxygen or Hypoxia is a hallmark of several tumors. In general, the p53 tumor suppressor is induced in conditions of severe hypoxia, whereby it induces apoptosis or cell death. However, some tumor cells can survive hypoxia by the activation of autophagy. This switch from apoptosis to autophagy resulting in tumor survival is not very well characterized, though several studies have implicated the intracellular localization of p53 as an important regulatory factor. In addition, mutations in p53 have also been associated with the activation of autophagic pathway. In this article, the authors characterize a p53 mutation, p53N236S and demonstrate that in response to hypoxia, it induces cell autophagy, represses apoptosis, and affects the intracellular localization of p53.

General Comments: The manuscript is well written. The data in general are supportive to the major conclusions, however at places (detailed below) western blots are difficult to interpret and the reader would benefit if quantification of blots is also included.

  • Does the mutation p53N236S, naturally occur in human cancers? Could the authors present an analysis from Cancer database showing the frequency of p53N236S occurrence in different human cancers? This would be helpful and will broaden the scope of the study.

Specific Comments:

  • Figure 1B: The western blot showing HIF-1 levels is not very clear and the gradual increase and reduction of HIF-1 at 16hrs is not obvious. It would be nice if the authors showed a quantification with statistics to confirm the HIF-1 levels. In addition, are the HIF-1 levels in the three cell types shown here comparable? As p53S/+ cells appear to have the least expression of HIF-1.
  • Figure 1D, 1F: Can the authors comment on why the p53S/+ cells do not show any reduction in apoptosis, while they show a significant reduction in PARP and Caspase 3 cleavage?
  • Line 242: Figure legend for D- please change B to “C”
  • Figure 2E: It would be nice if the authors could also present a statistical analysis.
  • For all the bar graphs: please check the labeling as some patterns do not match the indicated labels (example in Figure 3B) and hence is confusing for the reader.

Reviewer 2 Report

378 line ...is owever or however?

DFO blocks iron but what is happens if we have iron excess in this case of mutation?

Author Response

Comments and Suggestions for Authors

378 line ...is owever or however?

Response: We are very sorry for our mistake, we have corrected the typo.

DFO blocks iron but what is happens if we have iron excess in this case of mutation?

Response: Thanks for this question, as far as we understand, there are no reports of increased iron levels due to mutated p53, but perhaps we can test it later.

Round 2

Reviewer 1 Report

I am satisfied with the revised version and the authors' responses as they have addressed my concerns. However, I do have one suggestion: 1) Figure 2E: As per my suggestion, the authors added quantification and statistical analysis (figure 2F). But could they please clarify in figure legends, if they have done quantification for all 3 cell types (WT, p53S/+, p53S/S)? This still remains unclear.

Author Response

Response: Thanks for your carefully correction of our manuscript. The clarification of of 2F has been added in figure legend, and also the corrected panel order has been updated in the text.

This manuscript is a resubmission of an earlier submission. The following is a list of the peer review reports and author responses from that submission.